# CARE: Confidence-Aware Ratio Estimation for Medical Biomarkers

## Abstract

Ratio-based biomarkers – such as the proportion of necrotic tissue within a tumor – are widely used in clinical practice to support diagnosis and treatment planning. In automated clinical workflows, these biomarkers are typically estimated from segmentation outputs by computing region-wise ratios. However, the pointwise estimate captures no uncertainty measurement. To address this, we propose CARE, a *confidence-aware* ratio estimation framework considering the error propagation in the segmentation-to-biomarker pipeline. Specifically, we leverage tunable parameters to control the confidence level of the derived bounds. Experiments show that our method produces statistically sound confidence intervals, with tunable confidence levels, enabling more trustworthy application of predictive biomarkers in clinical workflows.

## 1 Introduction

Ratio-based biomarkers are widely utilized across various organs and imaging modalities as shown in Fig. 1a. For example, the necrosis-to-tumor ratio (NTR) [Henker et al., 2019, 2017] quantifies the proportion of necrotic (non-viable) tissue within a tumor. A straightforward method for computing these ratios involves using segmentation models to identify the subregion and the whole foreground region, and then calculating the ratio based on averaged softmax confidence scores over these regions. However, the interpretation of this point estimate can change once the confidence interval is considered, as illustrated in Fig. 1b. With a clinical threshold of 0.25 for initiating aggressive treatment, point estimates (case 1) alone suggest that Patient A would receive aggressive treatment (high ratio), whereas Patient B would receive mild treatment (low ratio). However, if the associated confidence interval spans the decision threshold (case 2), the estimation is flagged for mandatory expert review to mitigate potential misdiagnosis risk. Such double-check procedures are essential in clinical practice, as they provide an additional safeguard for patients and enhance the robustness of downstream decision-making.

To provide confidence measures for double-check, we propose CARE, the *first confidence-aware estimation framework specifically for ratio-based biomarkers*. CARE have several key advantages: i) **guaranteed coverage**, *i.e.*, the actual coverage probability of containing the true ratio is greater than the stated nominal confidence level; ii) instance-wise **adaptiveness**, *i.e.*, providing dynamic intervals that capture varying uncertainty degrees; iii) **tunable** confidence level with user-controlled tightness; iv) applicable as a **plug-in** module to any pretrained NN requiring neither architectural modifications nor training from scratch; v) computationally **efficient**, avoiding multiple sampling or repeated forward passes.

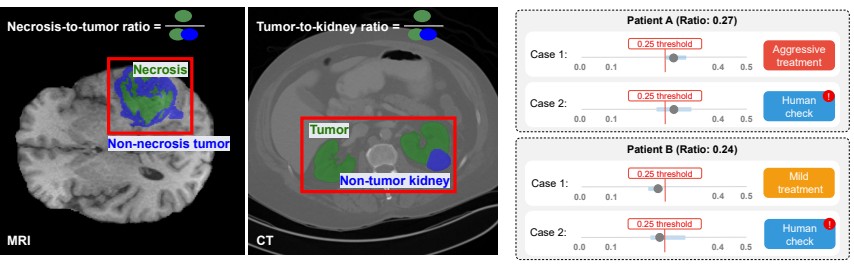

|(a) Biomarkers|(b) Clinical support|

Figure 1: **Medical background of ratio estimation and its role in clinical support. (a)**: Ratio-based biomarkers exist in many organs and modalities. **(b)**: An illustrative example. CARE calls for human check when the confidence interval crosses the predefined threshold.

## 2   CARE: Confidence-aware Ratio Estimation

The confidence intervals of CARE are constructed by combining two uncertainty sources by Boole's inequality [Boole, 1854, Dohmen, 2003]: i) an *estimation-based confidence interval* for the ratio estimator using Markov inequality [Resnick, 2003]; ii) a *calibration-based interval* to measure the prediction error from networks using conformal prediction [Shafer and Vovk, 2008].

**Proposition 2.1** (Estimation-based Confidence Interval). *Given an estimator $\hat{r} = \frac{\bar{y}}{\bar{x}}$ of the fraction $r = \frac{\mathbb{E}[y]}{\mathbb{E}[x]}$ with random variables $x$ and $y$, it holds with at least $1 - \alpha$ probability that*

$$r \in [\hat{r} - \beta_{r,\alpha}, \ \hat{r} + \beta_{r,\alpha}], \tag{1}$$

*where $\beta_{r,\alpha} := \frac{\sqrt{\mathrm{SE}_{\hat{r}}}}{\sqrt{\alpha}}$ as the bound's half-width, and $\mathrm{SE}_{\hat{r}} := \mathbb{E}\left[(\hat{r} - r)^2\right]$ as expected squared error.*

**Proposition 2.2** (Calibration-based Confidence Interval). *Consider a segmentation model $g(z) = (g_A(z), g_B(z))$ with the random variable $z$ representing pixel inputs of instance $I$, and targets $y_A$ and $y_B$. On a validation (calibration) set $\mathcal{D}_{cal}$, define $q_{A,\delta/2}$ and $q_{B,\delta/2}$ as the $1 - \delta/2$ quantile of the instance-wise volume bias or calibration errors of $g_A$ and $g_B$. Then, it holds with at least $1 - \delta$ probability that*

$$\frac{\mathbb{E}[y_A \mid I]}{\mathbb{E}[y_B \mid I]} \in \left[\frac{\mathbb{E}[g_A(z) \mid I]}{\mathbb{E}[g_B(z) \mid I]} - \epsilon_{l,\delta}, \frac{\mathbb{E}[g_A(z) \mid I]}{\mathbb{E}[g_B(z) \mid I]} + \epsilon_{u,\delta}\right], \tag{2}$$

*where $\epsilon_{l,\delta} := \frac{\mathbb{E}[g_A(z)]}{\mathbb{E}[g_B(z)]} - \frac{\mathbb{E}[g_A(z)] - q_{A,\delta/2}}{\mathbb{E}[g_B(z)] + q_{B,\delta/2}}$, $\epsilon_{u,\delta} := \frac{\mathbb{E}[g_A(z)] + q_{A,\delta/2}}{\mathbb{E}[g_B(z)] - q_{B,\delta/2}} - \frac{\mathbb{E}[g_A(z)]}{\mathbb{E}[g_B(z)]}$ are the widths of the lower and upper calibration bounds, respectively.*

Inspired by [Popordanoska et al., 2021], we offer two variants that allow clinicians to select either conservative or informative intervals. Specifically, informative CARE (V-Bias) takes the quantile of volume bias (|V-Bias|), and conservative CARE (ECE) considers ECE [Guo et al., 2017] quantiles. To combine both intervals, we make the following statement, which is analogous to multiple testing.

**Proposition 2.3** (Overall Confidence Interval). *Assume we have a ratio estimator $\hat{r} = \frac{\sum_i g_A(z_{i,I})}{\sum_i g_B(z_{i,I})}$ for pixel measurements $\{z_{i,I}\}_{i=1}^n$ of an instance $I$ based on neural network outputs $g(z_{i,I}) = (g_A(z_{i,I}), g_B(z_{i,I}))$. Let $y_A$ and $y_B$ be the instance-wise target random variables. Then, it holds with at least $1 - \alpha - \delta$ probability that*

$$\frac{\mathbb{E}[y_A \mid I]}{\mathbb{E}[y_B \mid I]} \in \left[\frac{\sum_i g_A(z_{i,I})}{\sum_i g_B(z_{i,I})} - \epsilon_{l,\delta} - \beta_{r,\alpha}, \frac{\sum_i g_A(z_{i,I})}{\sum_i g_B(z_{i,I})} + \epsilon_{u,\delta} + \beta_{r,\alpha}\right], \tag{3}$$

*where $\beta_{r,\alpha}$ is defined as in Prop. 2.1 and $\epsilon_{l,\delta}, \epsilon_{u,\delta}$ as in Prop. 2.2.*

The interval width $w = B_u - B_l$ measures the uncertainty level, as a result, a wide interval over thresholds alarms for manual examination. In experiments, we alternate through various $\alpha$ and $\delta$ for a fixed $\alpha + \delta$ with grid search to observe the impact on the interval width. This way, we can choose the smallest interval under a desired coverage rate.

# 3 Experiments

**Setup.** We evaluate CARE and Conformal Prediction on MSD-Task01 [Antonelli et al., 2022] with 4 segmentation models: nnUNet$_{2d, 3d}$ [Isensee et al., 2021], nnFormer [Zhou et al., 2021] and UNETR++ [Zhou et al., 2021]. The nested five-fold cross-validation is implemented: four folds for training (90%) and validation (10%), and the remaining one fold for testing.

Table 1: Comparison of the coverage guarantee on $C = 0.68$.

| Coverage (%) | nnUNet$_{2d}$ | nnUNet$_{3d}$ | nnFormer | UNETR++ |
|---|---|---|---|---|
| Conformal Prediction | $71.34_{\pm2.00}$ | $67.01_{\pm3.57}$ | $67.39_{\pm1.66}$ | $65.75_{\pm2.16}$ |
| CARE (V-Bias) | $93.61_{\pm1.14}$ | $86.60_{\pm1.49}$ | $81.92_{\pm1.31}$ | $76.43_{\pm2.21}$ |
| CARE (ECE) | $94.22_{\pm0.99}$ | $93.61_{\pm0.71}$ | $87.94_{\pm0.97}$ | $89.58_{\pm1.02}$ |

**Coverage guarantee.** We report coverage rate (%) at 0.68 confidence level in Table 1, which measures *the proportion of samples whose true values fall within the confidence intervals*. Empirically, our intervals show higher likelihoods of satisfying the prescribed confidence level of 0.68. We show more confidence thresholds on nnUNet$_{3d}$ in Fig. 2. Our method is flexibly tunable and consistently achieves coverage rates above the desired confidence levels.

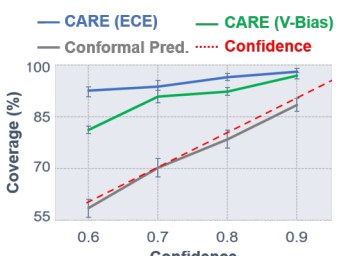

Figure 2: Coverage comparison across confidence levels.

**Adaptivity.** The confidence interval should be sample-adaptive to identify unreliable predictions effectively. We demonstrate this capability by examining the "dataset-level" interval distribution in Fig. 3. As observed, the results from Conformal Prediction lie within a narrow range and thus fail to effectively indicate which samples are unreliable. In contrast, CARE produces intervals that vary significantly in width. Given an interval width threshold, our method can effectively trigger alarms for cases with wide intervals (indicating high uncertainty), instead of giving uniformly narrow confidence ranges.

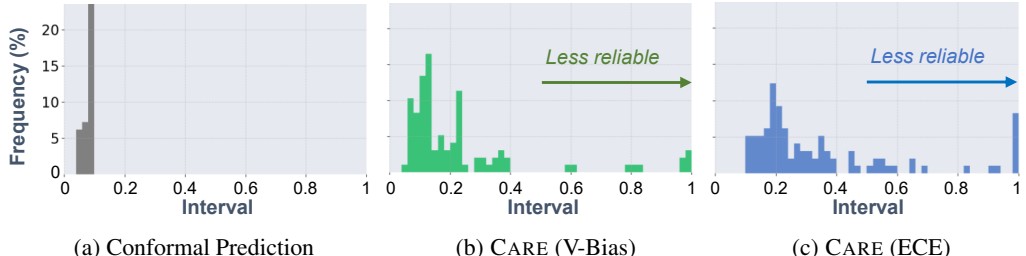

| (a) Conformal Prediction | (b) CARE (V-Bias) | (c) CARE (ECE) |

Figure 3: Comparison of interval distribution on $C = 0.68$. We report the frequency histogram of NTR intervals, where CARE triggers a human-check alarm for wide intervals.

# 4 Conclusion

We propose CARE, a confidence-aware framework for estimating ratio-based biomarkers from segmentation network outputs. Our method addresses a common limitation of prior works that focus solely on point estimates without confidence guarantees. We disentangle two key sources of uncertainty, *i.e.* network prediction error and statistical bias. Our framework offers several practical advantages: it operates as a model-agnostic plugin module, provides sample-level adaptive uncertainty estimates in a single forward pass without requiring multiple sampling, and allows users to flexibly adjust confidence levels. In summary, this work represents an important step toward trustworthy deployment of deep learning in clinical settings by providing practitioners with both accurate biomarker estimates and reliable confidence bounds.

## 5   Limitations and Acknowledgements

Despite the practical advantages, our work assumes that the validation and test sets are drawn from the same distribution. Although it is standard in supervised learning settings, but may not hold under domain shifts due to differences in scanners, acquisition protocols, or patient populations. As a result, our confidence interval may not remain valid in these scenarios. Addressing this challenge with label-free calibration error estimators (e.g. Wang et al. [2020], Popordanoska et al. [2024]) is a promising direction for future work.

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
