# OpenReview forum: "CARE: Confidence-aware Ratio Estimation for Medical Biomarkers"
_EurIPS.cc/2025/Workshop/MedEurIPS — EurIPS 2025 Workshop MedEurIPS Submission_

### Official Review · Reviewer_Exkm · 2025-10-30
**Overall a good paper with experimental utility for the novel confidence-aware ratio estimator.**

**Rating:** 7
**Confidence:** 4

**Review:**

**Summary**
- The paper introduces a confidence-aware framework for estimating ratio-based biomarkers from segmentation network output, a method that allows for the flexible adjustment of confidence levels.
Experimental results confirm that the framework outperforms baseline conformal prediction methods in both accuracy of interval calibration and adaptivity across diverse interval lengths. Overall, this paper is well-written and supported by clear experimental evidence.

**Suggestions:**
- An ablation study would be interesting to see the methods sensitivity to h various α and δ values
- Other uncertainty estimation could be used beyond conformal prediction (Ensemble, Bayesian ones).
- An experiment outlining the utility of uncertainty estimates compared to deterministic predictions (e.g. better performance under distribution shift or selective prediction task)

---

### Official Review · Reviewer_U22d · 2025-10-31
**Nice work that fits in the theme of Med meets NeurIPS**

**Rating:** 9
**Confidence:** 4

**Review:**

In this work, the authors propose a confidence-aware ratio estimation framework within the segmentation-to-biomarker pipeline. The task itself is highly interesting and represents a challenging medical problem that crucially requires probabilistic modeling with uncertainty estimation. I believe it will bring interesting discussion in the workshop.

---

### Decision · Program_Chairs · 2025-10-31

**Decision:**

Accept (Oral)

**Comment:**

Both reviewers praise the paper for addressing an important and timely problem at the intersection of uncertainty quantification and medical imaging.